# Implementation of Fault-Tolerant Encoding Circuit Based on Stabilizer Implementation and “Flag” Bits in Steane Code

**DOI:** 10.3390/e24081107

**Published:** 2022-08-11

**Authors:** Dongxiao Quan, Chensong Liu, Xiaojie Lv, Changxing Pei

**Affiliations:** 1School of Communication Engineering, Xidian University, Xi ’an 710071, China; 2Collaborative Innovation Center of Quantum Information of Shaanxi Province, Xidian University, Xi’an 710071, China

**Keywords:** quantum error correction, Steane code, stabilizer implementation, fault-tolerant encoding circuit design, “flag” bits

## Abstract

Quantum error correction (QEC) is an effective way to overcome quantum noise and de-coherence, meanwhile the fault tolerance of the encoding circuit, syndrome measurement circuit, and logical gate realization circuit must be ensured so as to achieve reliable quantum computing. Steane code is one of the most famous codes, proposed in 1996, however, the classical encoding circuit based on stabilizer implementation is not fault-tolerant. In this paper, we propose a method to design a fault-tolerant encoding circuit for Calderbank-Shor-Steane (CSS) code based on stabilizer implementation and “flag” bits. We use the Steane code as an example to depict in detail the fault-tolerant encoding circuit design process including the logical operation implementation, the stabilizer implementation, and the “flag” qubits design. The simulation results show that assuming only one quantum gate will be wrong with a certain probability *p*, the classical encoding circuit will have logic errors proportional to *p*; our proposed circuit is fault-tolerant as with the help of the “flag” bits, all types of errors in the encoding process can be accurately and uniquely determined, the errors can be fixed. If all the gates will be wrong with a certain probability *p*, which is the actual situation, the proposed encoding circuit will also be wrong with a certain probability, but its error rate has been reduced greatly from p to $p^{2}$ compared with the original circuit. This encoding circuit design process can be extended to other CSS codes to improve the correctness of the encoding circuit.

## 1. Introduction

By using superposition and entanglement, the ability of quantum computing will outperform that of classical computing in solving certain problems. However, quantum qubits are vulnerable to the environment, so they are fragile. Compared with classical errors, quantum errors are more complicated as there are not only bit errors but also phase errors that do not exist in classical computing. For the problem of errors in qubits, it is necessary to implement fault-tolerant quantum computing with the help of quantum error correction, which can identify and correct quantum errors. To implement quantum computing based on error-correcting codes, information qubits are firstly encoded into logical quantum qubits through the quantum encoding circuit, then the corresponding calculations are performed based on logical quantum gates, and finally the information qubits are obtained through the decoding circuit. Therefore, the encoding circuit is the first step of fault-tolerant quantum computing.

In 1995, Shor, based on the principle of information theory and quantum mechanics, reduced complex entangled state errors to a linear combination of X error and Z error on each qubit [1], and finally succeeded in constructing the first quantum error correction scheme using quantum repetition codes. He utilizes nine physical qubits to encode one logical information qubit, which is capable of correcting one error. In 1996, Steane proposed another quantum error correction scheme using seven physical qubits to encode one logical information qubit [2], which was named as Steane Code and became one of the most widely studied quantum error correction codes. In 1996, based on the idea of classical linear packet error correction codes, Calderbank, Shor, and Steane proposed the first system construction scheme of quantum error correction–the CSS code, which uses two special classical linear error correction codes [2,3]. The proposal of CSS code established the research theory of quantum error correction codes based on classical linear error correction codes, and more quantum error correction codes with better performance was being developed. In 1998, Bravyi and Kitaev introduced the concept of quantum topological code [4], which places physical qubits on a colored Latin lattice and each stabilizer is only related to a few qubits nearby. In 2005, the concept of quantum subsystem code [5] was introduced, which is a collection of multiple sub-encoding spaces of multiple qubits. In 2007, Ioffe and Mezard constructed asymmetric quantum BCH-LDPC codes based on classical BCH codes and low-density parity-check codes [6]. Asymmetric quantum error correction codes were further developed by a lot of researchers [7,8].

Due to the characteristics of nearest neighbors and high threshold, topological code attracted extensive attention, and gradually developed into famous quantum surface codes [9,10] and quantum color codes [11,12]. In 2016, Yoder and Kim proposed to achieve all Clifford gates by twisting the surface code and using lattice surgery [13]. A linear-time maximum likelihood decoder for surface codes over quantum erasure channel was proposed in 2017 [14]. Daniel Litinski and Felix von Oppen presented a planar surface-code-based scheme for fault-tolerant quantum computation where the overhead of single-qubit Clifford gates is significantly reduced [15]. In 2017, Sergey Bravyi et al. proposed an algorithm for simulating quantum error correction protocols based on two-dimensional surface code in the presence of coherent errors [16]. Darmawan and David proposed an efficient decoder for surface codes in 2018 [17]. In 2019, Christian Kraglund Andersen et al. initialized the cardinal states of the encoded logical qubit with an average logical fidelity of 96.1%, demonstrating the practicability of implementing quantum error correction in surface codes [18]. In 2020, Oscar Higgott et al. proposed a linear design for local surface code encoding [19], Fan Jihao et al. proposed asymmetric quantum tandem and tensor product codes [20]. Rui Chao et al. presented surface code error-correction schemes using only Pauli measurements on single qubits and pairs of nearest-neighbor qubits. They also developed minimized measurement sequences for syndrome extraction, enabling the improvement of logical error rate and fault-tolerance threshold [21]. In 2021, Marco Chiani et al. proposed the shortest codes with specified error correction capabilities according to the generalized quantum Hamming bound [22], Huang and Wu proposed a new construction of a nine-qubit error correction code, which is more suitable for high power qubit-flip noise [23]. J. F. Marques et al. realized a suite of logical operations on a distance-two logical qubit stabilized by repeated error detection cycles. This integration of high-fidelity logical operations with a scalable scheme for repeated stabilization is a milestone on the road to higher-distance superconducting surface codes [24].

The first step of using quantum error correction codes to protect information is the encoding circuit which encodes the information qubit into logical qubits. At present, there are mainly two kinds of quantum error correction encoding circuits. The first type is based on stabilizers measurement and correction. Fault-tolerant stabilizer measurement will cost a large number of qubits and quantum gates. During the stabilizers measurement, the stabilizers undergo random collapse, and the logical information is encoded later with the help of error correction operations. This kind of encoding method consumes more physical qubits, gates, and time slots, which increases the complexity and difficulty of implementation. The second category is based on stabilizer implementation, for example the classical encoding circuit for Steane code and nine-qubit Shor code. However, the existing encoding circuits have not considered the transmission of errors during the encoding process. A single error may propagate into multiple errors as the CNOT gates act between data qubits. For quantum error correction codes, we can only determine the errors according to the syndromes gotten from stabilizer measurements. However, some multi-qubit errors have the same syndromes as the single-qubit errors, they will be identified as single-qubit errors as single-qubit errors happen with a higher probability compared to multi-qubit errors. Because of the incorrect identification, the according correcting operation brings a logical error to the entire code word which can not be detected by stabilizer measurement following. In 2018, Chao and Reichardt proposed a fault-tolerant syndrome extraction method based on “flag” qubits, by the measurement results of “flag” qubits we can identify whether multi-qubit errors have occurred during the syndrome measurement process so that fault-tolerant syndrome extraction can be achieved [25]. In reference [26], a general fault-tolerant quantum error correction protocol using “flag” circuits for measuring stabilizers of arbitrary distance codes were put forward. In reference [27], “flag” qubits are used to realize fault-tolerant error correction for cyclic CSS codes. In reference [28], “flag” qubits are used to realize fault-tolerant quantum logic gates, so as to achieve fault-tolerant universal computation. In this paper, we introduce this idea to the encoding process, where “flag” qubits are used to mark whether multi-bit errors were introduced in the encoding process to enhance the fault tolerance of the encoding circuit.

In this paper, we firstly introduce the encoding circuit design process for CSS code based on stabilizer implementation and then use Steane code as an example to design the encoding circuit. Then based on the error propagating process of the CNOT gate, we analyze the un-fault tolerance of the original encoding circuit and introduce the “flag” qubits to identify multi-qubit errors. The analysis result shows that combining the syndromes of “flag” qubits and stabilizers, each error has a unique fingerprint and can therefore be identified and corrected exactly. The simulation result shows that the logical error rate is significantly reduced compared with the original circuit. This fault-tolerant encoding circuit design can be extended to other CSS codes to improve the correctness of the encoding circuit, which will facilitate the implementation of fault-tolerant quantum computing. The rest of this paper is organized as follows. Section 2 introduces the encoding circuit design process based on stabilizer implementation. Section 3 presents the error propagation process of the CNOT gate. Section 4 gives the fault-tolerant encoding circuit design process. Section 5 describes the simulation and results analysis. The last section concludes this paper.

## 2. Encoding Based on Stabilizer Implementation

For the encoding process based on stabilizers measurement and correction, fault-tolerant stabilizer measurement will cost a large number of qubits and quantum gates. Moreover, during the stabilizer measurement, the quantum states randomly collapse to positive or negative eigenstates of the stabilizer. For a QEC encoded block with *n* X type stabilizers, measuring the X type stabilizers will randomly get +1 or −1. The probability of getting all +1 syndrome is $1/2^{n}$, this is the only case where no fix operations are needed, and for all other cases fix operations are carried out to get the correct encoded logical states. So, this kind of encoding method consumes more physical qubits, gates, and time slots, which increases the complexity and difficulty of implementation.

This problem can be solved by designing an encoding method based on stabilizer implementation referring to the encoding formula. The logical quantum state of $|\bar{C_{1}}\bar{C_{2}}\cdots \bar{C_{k}}\rangle$$\left(C_{j}\in \left(0,1\right)\right)$ under the action of stabilizers can be written as Equation (Equation 1) [29,30], where *n* is the number of the physical qubits, *k* is the number of logical qubits, n−k is the number of stabilizers, $M_{i}$ is the *i*th stabilizer and $\bar{X_{i}}$ is the logical operation for the *i*th logical qubit:(1)$$\begin{array}{c} |\bar{C_{1}}\bar{C_{2}}\cdots \bar{C_{k}}\rangle ={\left(\frac{1}{\sqrt{2}}\right)}^{n-k}\prod_{i=1}^{n-k}\left(I+M_{i}\right){\bar{X_{1}}}^{C_{1}}{\bar{X_{2}}}^{C_{2}}\cdots {\bar{X_{k}}}^{C_{k}}|0_{1}0_{2}\cdots 0_{n}\rangle . \end{array}$$

If all the logical qubits are $|\bar{0}\rangle$, Equation (Equation 1) becomes Equation (Equation 2) as follows: (2)$$\begin{array}{c} |\bar{0_{1}}\bar{0_{2}}\cdots \bar{0_{k}}\rangle ={\left(\frac{1}{\sqrt{2}}\right)}^{n-k}\prod_{i=1}^{n-k}\left(I+M_{i}\right)|0_{1}0_{2}\cdots 0_{n}\rangle . \end{array}$$

For this formula, here we want to emphasize the condition it can be used. We need to choose the initial state ψ according to the logical *Z* operation $\bar{Z_{j}}$ and the *Z* type stabilizers $M_{iZ}$. Firstly it should satisfy
(3)$$\begin{array}{c} \bar{Z_{j}}|\bar{0_{1}}\bar{0_{2}}\cdots \bar{0_{k}}\rangle =|\bar{0_{1}}\bar{0_{2}}\cdots \bar{0_{k}}\rangle ,\forall j\in \left(1,2\cdots k\right), \end{array}$$
which means
(4)$$\begin{array}{c} {\left(\frac{1}{\sqrt{2}}\right)}^{n-k}\prod_{i=1}^{n-k}\left(I+M_{i}\right)\bar{Z_{j}}|\psi \rangle ={\left(\frac{1}{\sqrt{2}}\right)}^{n-k}\prod_{i=1}^{n-k}\left(I+M_{i}\right)|\psi \rangle ,\forall j\in \left(1,2\cdots k\right). \end{array}$$
So for a positive $\bar{Z_{j}}$ which contains *Z* operations on physical qubits, the physical bits involved in $\bar{Z_{j}}$ should be initialized to ∣0〉 or ∣1〉, and has an even number of ∣1〉; and the physical bits involved in $\bar{Z_{j}}$ should be initialized to ∣+〉 or ∣−〉 and has an even number of ∣−〉 if *X* operations are included in $\bar{Z_{j}}$. Moreover, for a positive *Z* type stabilizer $M_{iZ}$, the rule of initial state selection is consistent with the rule of $\bar{Z_{j}}$, and for a negative stabilizer $M_{iZ}$, the physical bits involved in this stabilizer should include an odd number of $|\bar{1}\rangle$ so as to make
(5)$$\begin{array}{c} \left(I+M_{iZ}\right)|\psi \rangle =|\psi \rangle . \end{array}$$
This means that, by the choice of the initial state, all the $\bar{Z_{j}}$ and $M_{iZ}$ are satisfied.

For the CSS code with positive $\bar{Z_{j}}$ and $M_{iZ}$, though there are many options for the initial state, usually, we choose ∣00⋯0〉 as the initial state, thus getting Equation (Equation 2). Moreover, as all the $M_{iZ}$ are satisfied by the choice of the initial state, so for the CSS code encoding only one logical qubit, $|\bar{0}\rangle$ can be expressed as:(6)$$\begin{array}{c} |\bar{0}\rangle ={\left(\frac{1}{\sqrt{2}}\right)}^{k_{x}}\prod_{i=1}^{k_{x}}\left(I+M_{iX}\right)|0_{1}0_{2}\cdots 0_{n}\rangle , \end{array}$$
where $k_{x}$ is the number of X type stabilizers.

Following, we realize the X type stabilizer to get $\left(I+M_{iX}\right)|\phi \rangle$ based on Hadamard gate and CNOT gates (Hadamard gate will be abbreviated as H gate for the rest of this paper) . Firstly, we choose one qubit *l* included in $M_{iX}$ meanwhile it is not entangled with other qubits in ∣ϕ〉, so ∣ϕ〉 can be written as
(7)$$\begin{array}{c} |\phi \rangle =|0_{l}\rangle |\varphi \rangle \end{array}$$
or
(8)$$\begin{array}{c} |\phi \rangle =|1_{l}\rangle |\varphi \rangle . \end{array}$$

If qubit *l* is ∣0〉, we act an H gate on qubit *l*, and then act a series of CNOT gates with qubit *l* as the control qubit and the other qubits included in $M_{iX}$ as the target qubits, so we can get
(9)$$\begin{array}{c} |\phi \rangle \overset{H_{l}}{\to }\left(|0_{l}\rangle +|1_{l}\rangle \right)|\varphi \rangle \overset{CNOT\cdots CNOT}{\to }|0_{l}\rangle |\varphi \rangle +M_{iX}|0_{l}\rangle |\varphi \rangle =\left(I+M_{iX}\right)|\phi \rangle . \end{array}$$

If qubit *l* is ∣1〉, we can add an extra Z gate after the H gate to revise the negative sign, and add an X gate on qubit *l* to realize the stabilizer, the process can be expressed as:(10)$$\begin{array}{c} \begin{array}{cc} |\phi \rangle & \overset{H_{l}}{\to }\left(|0_{l}\rangle -|1_{l}\rangle \right)|\varphi \rangle \overset{Z_{l}}{\to }\left(|0_{l}\rangle +|1_{l}\rangle \right)|\varphi \rangle \overset{CNOT\cdots CNOT}{\to }|0_{l}\rangle |\varphi \rangle +M_{iX}|0_{l}\rangle |\varphi \rangle \\ \;\; & \overset{X_{l}}{\to }|1_{l}\rangle |\varphi \rangle +M_{iX}|1_{l}\rangle |\varphi \rangle =\left(I+M_{iX}\right)|\phi \rangle . \end{array} \end{array}$$

After the implementation of stabilizer $M_{iX}$, the qubits involved in $M_{iX}$ will entangle with each other, they can not be chosen as the qubit to perform Hadamard operation (H operation) latter. Following, we can take $(I+M_{iX})|\phi \rangle$ as the initial state and realize the next stabilizer. In this way, we circulate this process until all the stabilizers are realized, so the coding of $|\bar{0}\rangle$ is realized.

We can see from Equation (Equation 1) that for the encoding of $|\bar{1}\rangle$, the initial state is $\bar{X_{1}}|00\cdots 0\rangle$. From Equations (9) and (10), we can find that when the qubit performing H operation is in a different state, the steps of stabilizer implementation are different, so in order to realize the encoding for α∣0〉+β∣1〉, the selected qubits should be in the same state for ∣00⋯0〉 and $\bar{X_{1}}|00\cdots 0\rangle$. Therefore, the shortest logical $\bar{X_{1}}$ should be selected, so as to ensure that there are qubits in a fixed state which could be selected for H operations to realize $M_{iX}$ latter. For a shortest $\bar{X_{1}}$, we choose one qubit *t* included in $\bar{X_{1}}$ as the information qubit, act as a series of CNOT gates with the information qubit as the control qubit and the other qubits included in $\bar{X_{1}}$ as the target qubits. Now the initial state ∣00⋯0〉 becomes
(11)$$\begin{array}{c} \alpha |0_{1}0_{2}\cdots 0_{n}\rangle +\beta \bar{X_{1}}|0_{1}0_{2}\cdots 0_{n}\rangle . \end{array}$$
The qubits not involved in $\bar{X_{1}}$ can be chosen as the qubit to perform H operations latter.

Following, we use Steane code as an example to show the encoding process. The X-stabilizer and Z-stabilizer of Steane code are
(12)$$\begin{array}{c} \begin{array}{cc} S_{1}^{X} & =X_{1}X_{3}X_{5}X_{7},\;S_{2}^{X}=X_{2}X_{3}X_{6}X_{7},\;S_{3}^{X}=X_{4}X_{5}X_{6}X_{7}\\ S_{1}^{Z} & =Z_{1}Z_{3}Z_{5}Z_{7},\;S_{2}^{Z}=Z_{2}Z_{3}Z_{6}Z_{7},\;S_{3}^{Z}=Z_{4}Z_{5}Z_{6}Z_{7}. \end{array} \end{array}$$

The logical operation for Steane code is usually written as $X_{L}=X_{1}X_{2}X_{3}X_{4}X_{5}X_{6}X_{7}$, here we need to use the shortest logical operation so we can obtain the equivalent logical operation by direct producing the logical operation and the stabilizers. In the circuit design, we choose $X_{L}^{\prime }=X_{1}X_{6}X_{7}$ as the logical operation which is gotten by $X_{L}⨂S_{2}^{X}⨂S_{3}^{X}$. We suppose the 1st qubit is the information qubit α∣0〉+β∣1〉 and other qubits are initialized to ∣0〉. We use $CNOT_{17}CNOT_{16}$ to implement the logical operation, where $CNOT_{ij}$ means a CNOT gate with the *i*th qubit as the control qubit and the jth qubit as the target qubit. Then for the stabilizer $S_{1}^{X}$, the 3rd or 5th qubit can be selected as the control qubits to perform H operation, here we choose $CNOT_{37}CNOT_{35}CNOT_{31}H_{3}$ to achieve $\left(I+S_{1}^{X}\right)$, where $H_{i}$ means a H gate on the *i*th qubit; similarly we use $CNOT_{27}CNOT_{26}CNOT_{23}H_{2}$ to achieve $\left(I+S_{2}^{X}\right)$ and $CNOT_{47}CNOT_{46}CNOT_{45}H_{4}$ to achieve $\left(I+S_{3}^{X}\right)$. Ultimately, Equation (Equation 13) can be obtained. The quantum circuit for encoding the information qubit into Steane code is shown in Figure 1:(13)$$\begin{array}{c} \begin{array}{c} |\bar{\varphi }\rangle =CNOT_{47}CNOT_{46}CNOT_{45}H_{4}CNOT_{27}CNOT_{26}CNOT_{23}H_{2}\\ CNOT_{37}CNOT_{35}CNOT_{31}H_{3}CNOT_{17}CNOT_{16}\left(\alpha |0_{1}\rangle +\beta |1_{1}\rangle \right)|0_{2}0_{3}\cdots 0_{n}\rangle . \end{array} \end{array}$$

The quantum circuit shown in Figure 1 is almost the same as the figure depicted in [31]. If we ignore the display positions of H gates and the implementation order of stabilizers, the only difference between the two circuits is that stabilizer $S_{2}^{X}$ is replaced by the equivalent stabilizer $X_{1}X_{2}X_{5}X_{6}$ which was gotten by $S_{2}^{X}⨂S_{1}^{X}$. Similarly, through the design process, the quantum circuits shown in [32,33] can all be obtained. All these circuits deformed the stabilizer so that each quantum bit is only used as the control bit or the target bit in the realization of stabilizers. These CNOT gates are all commutative and can be interchanged at will, which will benefit parallel computing. Following, we also deform the stabilizer to encode the quantum information.

## 3. Error Propagate Process of CNOT Gate

During the encoding process, a lot of CNOT gates are used to generate entanglement between the qubits to realize stabilizers, however, the errors may also propagate from one error to two errors by the CNOT gates. For the case an X error happened on the control qubit of CNOT gate, the X error will propagate to the target qubit as
(14)$$\begin{array}{c} \begin{array}{cc} CNOT\cdot X⨂I= & \left[\begin{array}{cccc} 1 & 0 & 0 & 0\\ 0 & 1 & 0 & 0\\ 0 & 0 & 0 & 1\\ 0 & 0 & 1 & 0 \end{array}\right]\cdot \left[\begin{array}{cccc} 0 & 0 & 1 & 0\\ 0 & 0 & 0 & 1\\ 1 & 0 & 0 & 0\\ 0 & 1 & 0 & 0 \end{array}\right]\\ = & \left[\begin{array}{cccc} 0 & 0 & 0 & 1\\ 0 & 0 & 1 & 0\\ 0 & 1 & 0 & 0\\ 1 & 0 & 0 & 0 \end{array}\right]\cdot \left[\begin{array}{cccc} 1 & 0 & 0 & 0\\ 0 & 1 & 0 & 0\\ 0 & 0 & 0 & 1\\ 0 & 0 & 1 & 0 \end{array}\right]=X⨂X\cdot CNOT; \end{array} \end{array}$$
and for the case a Z error happened on the target qubit of the CNOT gate, the Z error will propagate to the control qubit as
(15)$$\begin{array}{c} \begin{array}{cc} CNOT\cdot I⨂Z= & \left[\begin{array}{cccc} 1 & 0 & 0 & 0\\ 0 & 1 & 0 & 0\\ 0 & 0 & 0 & 1\\ 0 & 0 & 1 & 0 \end{array}\right]\cdot \left[\begin{array}{cccc} 1 & 0 & 0 & 0\\ 0 & -1 & 0 & 0\\ 0 & 0 & 1 & 0\\ 0 & 0 & 0 & -1 \end{array}\right]\\ = & \left[\begin{array}{cccc} 1 & 0 & 0 & 0\\ 0 & -1 & 0 & 0\\ 0 & 0 & -1 & 0\\ 0 & 0 & 0 & 1 \end{array}\right]\cdot \left[\begin{array}{cccc} 1 & 0 & 0 & 0\\ 0 & 1 & 0 & 0\\ 0 & 0 & 0 & 1\\ 0 & 0 & 1 & 0 \end{array}\right]=Z⨂Z\cdot CNOT. \end{array} \end{array}$$

The error propagation of the CNOT gate is shown in Figure 2, where an X error propagates from the control qubit to the target qubit and a Z error propagates from the target qubit to the control qubit. In the encoding circuit, as there are a lot of CNOT gates, so even if one error happened in the circuit, it may propagate to multiple errors, which cannot be corrected by Steane code. We will discuss these phenomena carefully and propose solutions in the next section.

## 4. Fault-Tolerant Encoding Circuit Design

### 4.1. Un-Fault Tolerance of the Original Encoding Circuit

For quantum error correction codes, we can only determine the errors based on the syndromes obtained from stabilizer measurements. The X errors can be determined by the Z-type syndromes and the Z errors can be determined by the X-type syndromes, we do not need to consider the Y errors as they can be divided into X errors and Z errors. Now we consider the errors of the circuit. For the CNOT gate, we consider six kinds of errors, $A_{i}^{CT},A\in \left(X,Z\right)$ means that both the control bit and target bit developed an A-type error at the *i*th CNOT gate, $A_{i}^{C},A\in \left(X,Z\right)\left(A_{i}^{T},A\in \left(X,Z\right)\right)$ references to that only an A-type error happened on the control$\left(target\right)$ bit at the *i*th CNOT gate. All the other errors of the CNOT gate can be obtained from the combination of these six types of errors. From Section 3 we know that an error before the CNOT gate, no matter whether it is generated by the storage or the quantum gate, can be equivalent to one type of error of the CNOT gate, so in the following analysis, we only consider the errors of CNOT gate. Furthermore, the first two CNOT gates are used to realize the logical operation and are similar to repeated encoding on the Z basis. The Z error on the information qubit can be seen as the information has been changed, so it can not be detected later. Such a problem also exists in the encoding circuit based on stabilizer measurement, so in this manuscript, we suppose the first two CNOT gates are perfect, we only consider the errors of CNOT gates for stabilizer implementation.

Here we deform the stabilizer and also change the stabilizer implementation order so as to keep consistent with Equation (Equation 12). We firstly add the Z-type stabilizer measurement to obtain Figure 3, and the syndromes are written as $S_{1Z},S_{2Z},S_{3Z}$, where $S_{iZ}=M\left(S_{i}^{Z}\right)$ is the measurement result according to the Z-type stabilizer. The syndrome measurements shown in Figure 3 are not fault-tolerant, in actual implementation, the syndrome measurements based on “cat state” [34,35,36], encoding blocks [37,38] and ”flag” qubit [25,26,27,28] can be adopted to ensure fault-tolerance.

We suppose only one error happened during the 9 CNOT gates labeled as 1 to 9 from left to right for stabilizer implementation, we can get Table 1. For all the single qubit errors, they can be corrected exactly according to the syndromes. For the first column, if there is an $X_{i}^{CT}$ error happened on the first CNOT gate for every stabilizer implementation, it will introduce an extra stabilizer to the quantum code, so there will be no error, in fact, and no fix operations are needed. For the three-qubit errors, for example, when there is an $X_{1}^{C}$ error or $X_{2}^{CT}$ error, both will cause an $X_{3}X_{5}X_{7}$ error. The $X_{3}X_{5}X_{7}$ error has the same syndrome as $X_{1}$ errors, and an $X_{1}$ error is more likely to occur, so it will be considered as the $X_{1}$ error and corrected by $X_{1}$. In total, all introduced operations to the quantum code are the stabilizer $X_{1}X_{3}X_{5}X_{7}$, which has no influence on the code. We call $X_{3}X_{5}X_{7}$ and $X_{1}$ are equivalent errors. $X_{1}$ and $X_{2}X_{5}X_{6}$, $X_{5}$ and $X_{4}X_{6}X_{7}$ also belong to this case. For the above cases, we can encode correctly. However, for the case when there is an $X_{2}^{C}$ or an $X_{3}^{CT}$ error, the introduced error $X_{3}X_{7}$ will be considered as and corrected by $X_{4}$ as they have the same syndrome, and the latter happens with large probability. After the correction, all introduced operations to the quantum code are $X_{3}X_{4}X_{7}=X_{L}⨂S_{1}^{X}⨂S_{2}^{X}$, which is a logical X operation to the code, so we will get the wrong quantum code and the error can not be detected by syndrome measurement anymore. This situation also occurs in $X_{5}^{C}$,$X_{6}^{CT}$,$X_{8}^{C}$, and $X_{9}^{CT}$, which are shown in red in the table.

Similarly, the same problem exists for Z errors. For the same encoding circuit, we add the X-type stabilizer measurement to obtain Figure 4, and the syndromes are written as $S_{1X},S_{2X},S_{3X}$, where $S_{iX}=M\left(S_{i}^{X}\right)$ is the measurement result according to the X-type stabilizer. We can get the relation of the CNOT gate errors, the errors introduced into the code block, and the syndromes of the errors as shown in Table 2.

Similarly, we analyze these errors and find out the errors that may introduce additional logical operations to the code block. For the $Z_{2}^{CT}$ error, it will introduce $Z_{2}Z_{3}Z_{4}Z_{5}$ to the code block. It is easily found that $Z_{2}Z_{3}Z_{4}Z_{5}$ is a stabilizer which is equivalent to $Z_{2}Z_{3}Z_{6}Z_{7}⨂Z_{4}Z_{5}Z_{6}Z_{7}$, so it has no influence to the code block. For all the single qubit errors, they can be exactly corrected according to the syndromes. For the two-qubit errors, they will be recognized as the single qubit errors in the same column, and brings additional logical operation to the code block. Take the error $Z_{1}^{T}=Z_{1}$ as an example to show the process in detail. As shown in Figure 2, the Z error propagates from the target qubit to the control qubit, qubit 1 is the target qubit of CNOT4, so, in the end, the total errors in the data block are $Z_{1}Z_{2}$. $Z_{1}Z_{2}$ error has the same syndrome as $Z_{3}$, so it will be corrected by $Z_{3}$, and the total error to the code is $Z_{1}Z_{2}Z_{3}$, which is a logical Z operation to the code. The two qubit errors shown in red in Table 2 all belong to this case. For the error $Z_{2}Z_{4}Z_{5}$, it is equivalent to $Z_{3}$, so it can be corrected based on the syndrome. The errors shown in red in the first column are equivalent as they are different logical Z operations, which also need to be distinguished by other means.

In fact, for both the *X* and *Z* errors in the same column, they will all be recognized as and corrected by the shortest error in the column. If the error is equivalent to the shortest error, it can be corrected, or the wrong correction will bring a logical operation to the code. The errors that need to be further distinguished are the high-weight errors generated by CNOT gates. For such a problem, the idea of “flag”-based syndrome extraction is introduced, and errors will be distinguished by adding ancillary qubits.

### 4.2. “flag”-Based Syndrome Measurement

The basic idea of a fault-tolerant encoding circuit is to use “flag” qubits to capture high-weight errors. The qubits in the encoding circuit that need to be supervised are marked by adding ancillary qubits. The errors are then determined by combining the “flag” qubits with the stabilizer measurements.

From Section 3 we know that there are two cases that will cause high-weight errors and need to be monitored. The first case is the X error on the control qubit which may transmit to other qubits. For this case, we can use Figure 5 to monitor whether an X error happened on the control qubit. In this circuit, two CNOT gates with the data qubit need to be monitored as the control qubit and the “flag” qubit as the target qubit are added. By the measurement result of the “flag” qubit we can determine whether an X error happened on the control qubit between the two added CNOT gates.

Firstly, two CNOT gates with the same control qubit are commutative as
(16)$$\begin{array}{c} \begin{array}{cc} \; & CNOT_{1,2}\cdot CNOT_{1,3}\\ = & \left[\begin{array}{cccccccc} 1 & 0 & 0 & 0 & 0 & 0 & 0 & 0\\ 0 & 1 & 0 & 0 & 0 & 0 & 0 & 0\\ 0 & 0 & 1 & 0 & 0 & 0 & 0 & 0\\ 0 & 0 & 0 & 1 & 0 & 0 & 0 & 0\\ 0 & 0 & 0 & 0 & 0 & 0 & 1 & 0\\ 0 & 0 & 0 & 0 & 0 & 0 & 0 & 1\\ 0 & 0 & 0 & 0 & 1 & 0 & 0 & 0\\ 0 & 0 & 0 & 0 & 0 & 1 & 0 & 0 \end{array}\right]\cdot \left[\begin{array}{cccccccc} 1 & 0 & 0 & 0 & 0 & 0 & 0 & 0\\ 0 & 1 & 0 & 0 & 0 & 0 & 0 & 0\\ 0 & 0 & 1 & 0 & 0 & 0 & 0 & 0\\ 0 & 0 & 0 & 1 & 0 & 0 & 0 & 0\\ 0 & 0 & 0 & 0 & 0 & 1 & 0 & 0\\ 0 & 0 & 0 & 0 & 1 & 0 & 0 & 0\\ 0 & 0 & 0 & 0 & 0 & 0 & 0 & 1\\ 0 & 0 & 0 & 0 & 0 & 0 & 1 & 0 \end{array}\right]\\ = & \left[\begin{array}{cccccccc} 1 & 0 & 0 & 0 & 0 & 0 & 0 & 0\\ 0 & 1 & 0 & 0 & 0 & 0 & 0 & 0\\ 0 & 0 & 1 & 0 & 0 & 0 & 0 & 0\\ 0 & 0 & 0 & 1 & 0 & 0 & 0 & 0\\ 0 & 0 & 0 & 0 & 0 & 1 & 0 & 0\\ 0 & 0 & 0 & 0 & 1 & 0 & 0 & 0\\ 0 & 0 & 0 & 0 & 0 & 0 & 0 & 1\\ 0 & 0 & 0 & 0 & 0 & 0 & 1 & 0 \end{array}\right]\cdot \left[\begin{array}{cccccccc} 1 & 0 & 0 & 0 & 0 & 0 & 0 & 0\\ 0 & 1 & 0 & 0 & 0 & 0 & 0 & 0\\ 0 & 0 & 1 & 0 & 0 & 0 & 0 & 0\\ 0 & 0 & 0 & 1 & 0 & 0 & 0 & 0\\ 0 & 0 & 0 & 0 & 0 & 0 & 1 & 0\\ 0 & 0 & 0 & 0 & 0 & 0 & 0 & 1\\ 0 & 0 & 0 & 0 & 1 & 0 & 0 & 0\\ 0 & 0 & 0 & 0 & 0 & 1 & 0 & 0 \end{array}\right]\\ = & CNOT_{1,3}\cdot CNOT_{1,2}, \end{array} \end{array}$$
the second CNOT gate can change order with the first or the third CNOT gate and two same adjacent CNOT gates will get *I*. So if there is no error, the “flag” qubit and these two additional CNOT gates have no effect on the original circuit. From Section 3, we can obtain if there is an X error that happened before the first CNOT gate, the “flag” qubit will be flipped twice and will have no errors; if there is an X error that happened between the first and the third CNOT gates, the “flag” qubit will be flipped and we can detect this error by the measurement of the “flag” qubit. Here we ignore the errors of the data qubits as we will combine the results of the “flag” qubits and the syndromes of the stabilizer to determine the error latter.

The second case of high-weight errors is the Z error on the target qubit which may transmit to other qubits by the following CNOT gates. Similarly, we can use Figure 6 to monitor whether a Z error happened on the target qubit, where two additional CNOT gates with the “flag” qubit as the control qubit and the data qubit need to be monitored as the target qubit are added. By the measurement result of the “flag” qubit we can determine whether a Z error happened on the target qubit between the two added CNOT gates.

Similarly, two CNOT gates with the same target qubit are commutative as
(17)$$\begin{array}{c} \begin{array}{cc} \; & CNOT_{1,2}\cdot CNOT_{3,2}\\ = & \left[\begin{array}{cccccccc} 1 & 0 & 0 & 0 & 0 & 0 & 0 & 0\\ 0 & 1 & 0 & 0 & 0 & 0 & 0 & 0\\ 0 & 0 & 1 & 0 & 0 & 0 & 0 & 0\\ 0 & 0 & 0 & 1 & 0 & 0 & 0 & 0\\ 0 & 0 & 0 & 0 & 0 & 0 & 1 & 0\\ 0 & 0 & 0 & 0 & 0 & 0 & 0 & 1\\ 0 & 0 & 0 & 0 & 1 & 0 & 0 & 0\\ 0 & 0 & 0 & 0 & 0 & 1 & 0 & 0 \end{array}\right]\cdot \left[\begin{array}{cccccccc} 1 & 0 & 0 & 0 & 0 & 0 & 0 & 0\\ 0 & 0 & 0 & 1 & 0 & 0 & 0 & 0\\ 0 & 0 & 1 & 0 & 0 & 0 & 0 & 0\\ 0 & 1 & 0 & 0 & 0 & 0 & 0 & 0\\ 0 & 0 & 0 & 0 & 1 & 0 & 0 & 0\\ 0 & 0 & 0 & 0 & 0 & 0 & 0 & 1\\ 0 & 0 & 0 & 0 & 0 & 0 & 1 & 0\\ 0 & 0 & 0 & 0 & 0 & 1 & 0 & 0 \end{array}\right]\\ = & \left[\begin{array}{cccccccc} 1 & 0 & 0 & 0 & 0 & 0 & 0 & 0\\ 0 & 0 & 0 & 1 & 0 & 0 & 0 & 0\\ 0 & 0 & 1 & 0 & 0 & 0 & 0 & 0\\ 0 & 1 & 0 & 0 & 0 & 0 & 0 & 0\\ 0 & 0 & 0 & 0 & 1 & 0 & 0 & 0\\ 0 & 0 & 0 & 0 & 0 & 0 & 0 & 1\\ 0 & 0 & 0 & 0 & 0 & 0 & 1 & 0\\ 0 & 0 & 0 & 0 & 0 & 1 & 0 & 0 \end{array}\right]\cdot \left[\begin{array}{cccccccc} 1 & 0 & 0 & 0 & 0 & 0 & 0 & 0\\ 0 & 1 & 0 & 0 & 0 & 0 & 0 & 0\\ 0 & 0 & 1 & 0 & 0 & 0 & 0 & 0\\ 0 & 0 & 0 & 1 & 0 & 0 & 0 & 0\\ 0 & 0 & 0 & 0 & 0 & 0 & 1 & 0\\ 0 & 0 & 0 & 0 & 0 & 0 & 0 & 1\\ 0 & 0 & 0 & 0 & 1 & 0 & 0 & 0\\ 0 & 0 & 0 & 0 & 0 & 1 & 0 & 0 \end{array}\right]\\ = & CNOT_{3,2}\cdot CNOT_{1,2}. \end{array} \end{array}$$
If there is no error, the “flag” qubit and these two additional CNOT gates have no effect on the original circuit. From Section 3, we can obtain if there is a Z error that happened on qubit 2 between the first and the third CNOT gate, it will introduce a Z operation on the “flag” qubit, which will be detected by the measurement of the “flag” qubit in X basis.

From the above analysis, we can use Figure 5 to detect the X error on the control qubit and Figure 6 to detect the Z error on the target qubit. We can not monitor the Z error on the control bit, such as the information bit when realizing the logical operation. We need to deform the stabilizer in the encoding circuit design to make sure the qubit is only used as the control qubit or as the target qubit. By analyzing Table 1 and Table 2, we can choose some positions to add the “flag” qubits, the inequivalent errors with the same syndromes can be distinguished, so that fault-tolerant encoding can be realized.

### 4.3. Fault-Tolerant Encoding Circuit

Firstly, we select the locations to add “flag” qubits to distinguish the inequivalent X-type errors. In Table 1, the errors shown in red are the errors that need to be distinguished. We can use Figure 5 to monitor the X type errors on the control qubits of CNOT2, CNOT3, CNOT5, CNOT6, CNOT8, CNOT9, so we can get Figure 7. In this figure, we do not distinguish the control qubit errors for the same stabilizers, as they can be further distinguished by the syndromes of stabilizers. The possible errors of the circuit and the syndromes of the flag qubits and the stabilizers are shown in Table 3.

From Table 3, we can see if the control qubit for the *i*th stabilizer occurs, an X error, the measurement of the *i*th flag qubit will get 1, so the syndromes of the “flag” qubit can reflect whether the control bits have an X error. Any cell in the table corresponds to a unique error, therefore the exact error can be determined by the overall syndromes of flag qubits and Z stabilizers and thus can be accurately corrected.

Similarly, for the inequivalent Z errors shown in red in Table 2, we can use Figure 6 to monitor the Z errors on the target qubits of CNOT1, CNOT4,CNOT5, CNOT7, CNOT6, CNOT8, CNOT3, CNOT9, so Figure 8will be obtained. In this figure, adjacent CNOT gates with the same target qubit share one “flag” qubits, the exact error can be further distinguished by the syndromes of stabilizers. The possible errors of the circuit and the syndromes of the “flag” qubits and the stabilizers are shown in Table 4.

From Table 4, we can see if an Z error happened on qubit 1, the measurement of “flag” 4 will get 1; if an Z error happened on qubit i(i=5,6,7), the measurement of “flag” *i* will get 1. So the syndromes of the “flag” qubits can reflect whether the target bits have a Z error. Any cell in the table corresponds to a unique error, which means unique identification is also achieved for Z-errors, therefore the exact error can be determined by the overall syndromes of the “flag” qubits and X stabilizers and thus can be accurately corrected.

As both the X error and Z error may occur during the encoding process, so combining Figure 7 and Figure 8 gives a complete fault-tolerant Steane code encoding circuit. In the encoding process, three auxiliary qubits are used to monitor the X errors and four extra auxiliary qubits are added for monitoring the Z errors, the total encoding circuit requires seven auxiliary qubits. If all the “flag” qubits are 0, it indicates that there is no high-weight error that happened in the encoding process. Following the encoding circuit, we can add one round of stabilizer measurement as shown in Figure 3 and Figure 4 to determine the errors exactly. We don’t need to worry about the seven extra bits used in the encoding process, because they can be initialized again for subsequent stabilizer measurements.

## 5. Simulation and Analysis

In this section, we use Qiskit from IBM to simulate the performance of the original encoding circuit shown in in [31] and our proposed encoding circuit combined by Figure 7 and Figure 8. For the encoding circuit, as the storage error is much smaller than the quantum gate error, so here we neglect the storage error. We also neglect the errors of the CNOT gates connected with the ancillary qubits, as one ancillary qubit only acts with one data qubit, and it will at most introduce one error to the data block. Moreover, both these two kinds of errors can be equivalent to a kind of quantum gate error, so ignoring these errors will not affect the fault-tolerance analysis of the circuit. Here we mainly consider the Hadamard errors and CNOT errors. Suppose the quantum gate occurs an error with the same probability *p*. The Hadamard error means attempting to perform a Hadamard operation *H*, but performing, in addition, one of the single qubit operations *X*, *Y*, or *Z*, each with probability p/3. The CNOT error means attempting to perform a CNOT operation, but instead performing in addition one of the two-qubit operations I⨂X, I⨂Z, I⨂Y, X⨂I, X⨂X, X⨂Z, X⨂Y, Z⨂I, Z⨂X, Z⨂Z, Z⨂Y, Y⨂I, Y⨂X, Y⨂Z, Y⨂Y, each with probability p/15, where A⨂B means after the CNOT gate, the operation *A* acts on the control qubit and operation *B* acts on the target qubit.

The simulation circuit begins with the encoding circuit and is then followed by the syndrome measurement circuit. Error correction is carried out based on all the syndromes and followed by the decoding circuit which is reciprocal to the encoding circuit. We then compare the output information with the input information to calculate the logical error rate. For the simulation, we suppose two cases. Firstly, for all the three H gate and nine CNOT gates, every time we choose only one gate, and suppose its error probability is *p*. We run the simulation enough times to count the logic error rate. The results for the classical circuit are shown in Figure 9, there will be no logical error for the H gate error. The theoretical error rate for all the nine CNOT gates is shown in solid lines with different colors, the simulation results are marked with different shapes. From the figure, we can see the simulation results are consistent with the theoretical analysis as shown in Table 1 and Table 2. The logical error rate is proportional to the error rate of each gate, and the 3rd and 6th CNOT gates are more likely to cause logical errors with a probability of 2p/3, and the 2nd, 4th, and 7th CNOT gates will cause logical errors with the smallest probability 4p/15. If we randomly choose one of the H gates and CNOT gates to occur one error, the logical error rate is shown in purple which is also proportional to the error rate of each gate, obviously, this error rate is unacceptable. Though we suppose only one error happens, however, there will also be logical errors as the circuit is not fault-tolerant. For our proposed circuit, there is no logical error under this assumption.

In practical application, we can not guarantee that only one gate has an error. So here for the second case, we suppose all the three H gates and nine CNOT gates occur an error with the same probability p. For different probability p, we run the simulation enough times to count the logic error rate. The results for the two circuits are shown in Figure 10. From which we can see the logical error rate of the classical circuit is bigger than that of the quantum gate, which means using the classical circuit can not protect the information. While for the proposed circuit its logical error rate is significantly reduced, as its logical error rate is $\propto p^{2}$. The threshold of our proposed circuit is 0.077, when the quantum gate error rate is smaller than this value, we can get a better result using this circuit. The smaller the error rate of quantum gates, the greater the improvement of the fidelity of logical quantum states. The logical error rate for the proposed circuit is also not ideal for large p as the code distance is only three, which can only correct all the single-qubit errors and some two-qubit errors. We can reduce the logic error probability by using error correcting codes with larger code distance.

Although seven auxiliary qubits are used in the encoding process, these auxiliary qubits can be used for subsequent stabilizer measurements by resetting, and the number of physical quantum qubits needed for the whole system does not increase. Compared with the encoding method of stabilizers measurement, this encoding method is easier to implement due to its relatively low complexity and low resource consumption. The fault-tolerant encoding circuit can be extended to quantum surface codes, which is useful for the optimization and applications of surface codes.

## 6. Conclusions

In this paper, we firstly analyze the encoding circuit design process for CSS code based on stabilizer implementation and then use Steane code as an example to design the encoding circuit. Through the design process, the classic encoding circuit and other variant circuits can be obtained, which shows the correctness of the encoding circuit design process. Secondly, we theoretically analyze the error propagation principle of the CNOT gate, and on this basis, we analyze the reasons why a typical encoding circuit is not fault-tolerant. Following, we give the principle of how to add “flag” qubits to detect the high-weight errors and apply it to the classical encoding circuit for Steane code. Theoretical analysis shows that any inequivalent error will get a unique syndrome, so it can be accurately distinguished and corrected. At last, we simulate the performance of the circuit, the result shows that if at most only one error happens in the encoding process, our proposed circuit can get the correct encoding result based on the error-correction process while the original circuit will get the wrong result with a certain probability. If it is assumed that every gate occurs an error with a certain probability, which is also the actual situation, the proposed encoding circuit also has a certain probability of error, but its error rate has been reduced greatly from p to $p^{2}$ compared with the original circuit. These simulations show the effectiveness and superiority of the proposed method. In the future work, we can fully consider the characteristics of the stabilizers of surface code and extend this method to surface code, which is the most potential error correcting code in quantum computing.

## Figures and Tables

**Figure 1 entropy-24-01107-f001:**
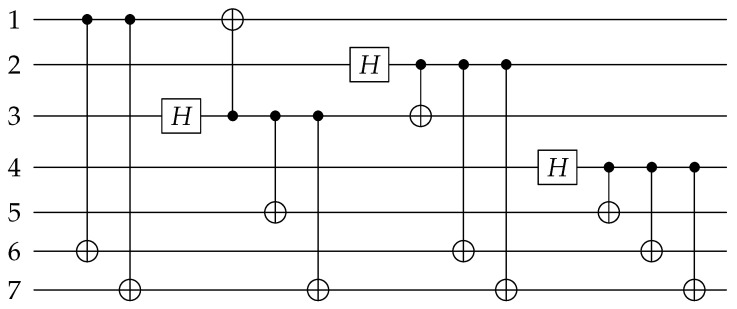
Quantum circuit for encoding the information qubit into Steane code.

**Figure 2 entropy-24-01107-f002:**
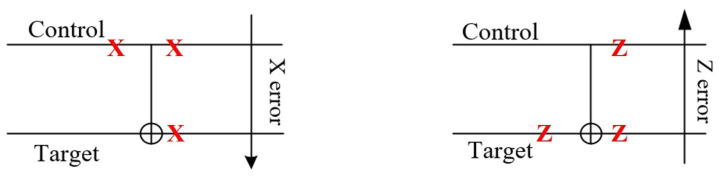
Schematic diagram of the error propagate of CNOT gate, where an X error propagates from the control qubit to the target qubit and a Z error propagates from the target qubit to the control qubit.

**Figure 3 entropy-24-01107-f003:**
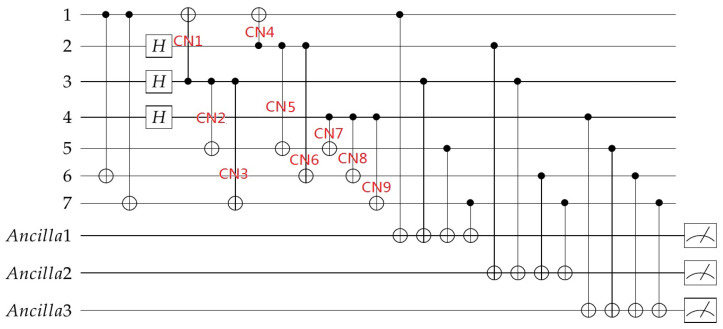
Quantum circuit for encoding the information qubit into Steane code with Z-type stabilizer measurement.

**Figure 4 entropy-24-01107-f004:**
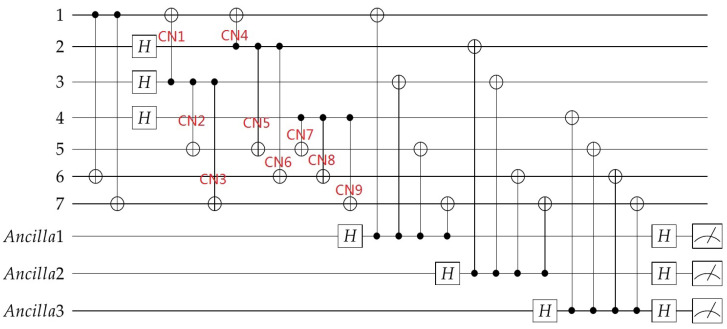
Quantum circuit for encoding the information qubit into Steane code with X-type stabilizer measurement.

**Figure 5 entropy-24-01107-f005:**
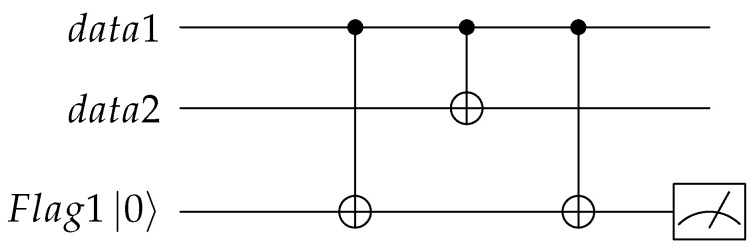
Using the “flag” qubit to detect the X error on the control qubit.

**Figure 6 entropy-24-01107-f006:**
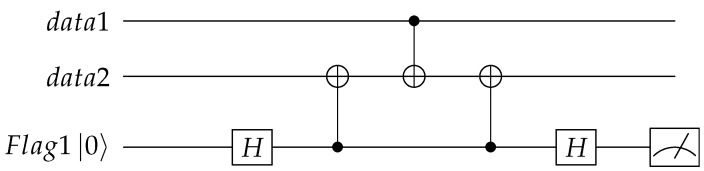
Using the “flag” qubit to detect the Z error on the target qubit.

**Figure 7 entropy-24-01107-f007:**
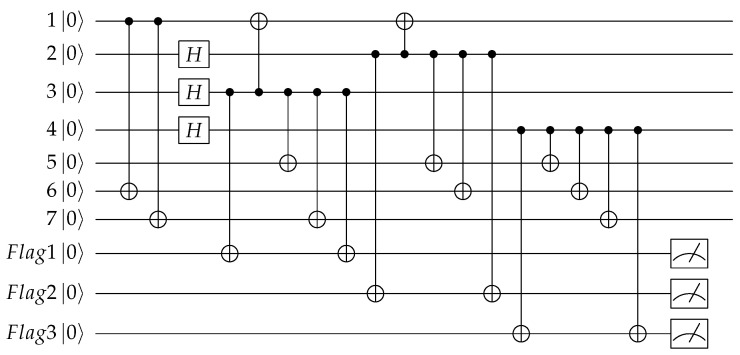
Quantum circuit for encoding the information qubit into Steane code with “flag” qubits to monitor the high-weight X errors.

**Figure 8 entropy-24-01107-f008:**
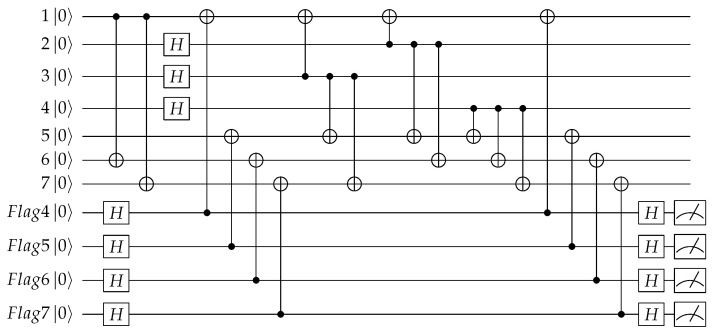
Quantum circuit for encoding the information qubit into Steane code with “flag” qubits to monitor the high-weight Z errors.

**Figure 9 entropy-24-01107-f009:**
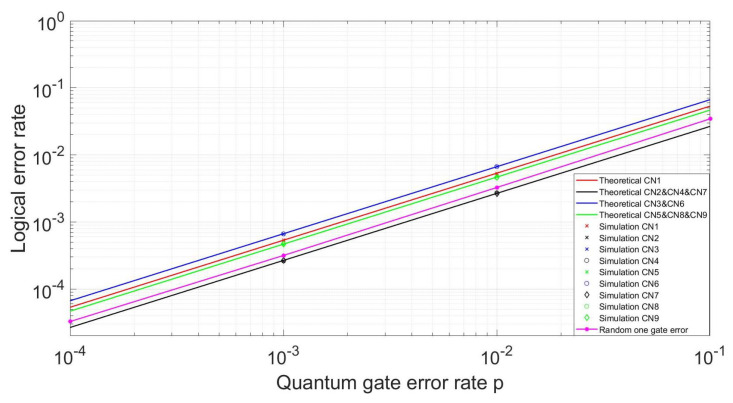
Logical error rates for the classical encoding circuit.

**Figure 10 entropy-24-01107-f010:**
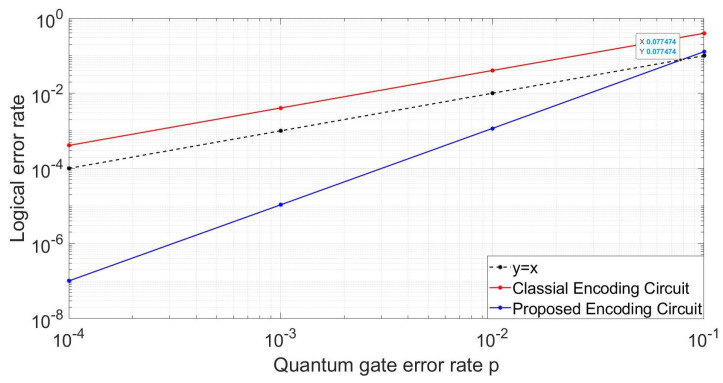
Comparison of the logical error rates for the classical encoding circuit and our proposed encoding circuit.

**Table 1 entropy-24-01107-t001:** The relation of possible errors of CNOT gates, the according qubit errors they generated, and the Z-type syndromes will get.

$S_{1Z},S_{2Z},S_{3Z}$	000	001	010	011	100	101	110	111
CNOT error	$X_{1}^{CT}$	$X_{2}^{C}=X_{3}^{CT}$	$X_{6}^{C}$	$X_{6}^{T}=X_{8}^{T}$	$X_{1}^{T}=X_{4}^{T}$	$X_{2}^{T}=X_{5}^{T}=X_{7}^{T}$	$X_{3}^{C}$	$X_{3}^{T}=X_{9}^{T}$
qubit error	$X_{1}X_{3}X_{5}X_{7}$	$X_{3}X_{7}$	$X_{2}$	$X_{6}$	$X_{1}$	$X_{5}$	$X_{3}$	$X_{7}$
CNOT error	$X_{4}^{CT}$	$X_{9}^{C}$	—	—	$X_{1}^{C}=X_{2}^{CT}$	$X_{7}^{C}=X_{8}^{CT}$	$X_{8}^{C}=X_{9}^{CT}$	—
qubit error	$X_{1}X_{2}X_{5}X_{6}$	$X_{4}$	—	—	$X_{3}X_{5}X_{7}$	$X_{4}X_{6}X_{7}$	$X_{4}X_{7}$	—
CNOT error	$X_{7}^{CT}$	$X_{5}^{C}=X_{6}^{CT}$	—	—	$X_{4}^{C}=X_{5}^{CT}$	—	—	—
qubit error	$X_{4}X_{5}X_{6}X_{7}$	$X_{2}X_{6}$	—	—	$X_{2}X_{5}X_{6}$	—	—	—

**Table 2 entropy-24-01107-t002:** The relation of possible errors of CNOT gates, the according qubit errors they generated, and the X-type syndromes will get.

$S_{1X},S_{2X},S_{3X}$	000	001	010	011	100	101	110	111
CNOT error	$Z_{1}^{CT}$	$Z_{7}^{C}=Z_{8}^{C}=Z_{9}^{C}$	$Z_{4}^{C}=Z_{5}^{C}=Z_{6}^{C}$	$Z_{8}^{T}$	$Z_{4}^{T}$	$Z_{7}^{T}$	$Z_{1}^{T}=Z_{4}^{CT}$	$Z_{9}^{T}$
qubit error	$Z_{1}Z_{2}Z_{3}$	$Z_{4}$	$Z_{2}$	$Z_{6}$	$Z_{1}$	$Z_{5}$	$Z_{1}Z_{2}$	$Z_{7}$
CNOT error	$Z_{2}^{CT}$	—	$Z_{6}^{T}=Z_{8}^{CT}$	—	$Z_{5}^{T}=Z_{7}^{CT}$	—	$Z_{1}^{C}=Z_{2}^{C}=Z_{3}^{C}$	—
qubit error	$Z_{2}Z_{3}Z_{4}Z_{5}$	—	$Z_{4}Z_{6}$	—	$Z_{4}Z_{5}$	—	$Z_{3}$	—
CNOT error	$Z_{3}^{CT}$	—	—	—	—	—	$Z_{3}^{T}=Z_{9}^{CT}$	—
qubit error	$Z_{3}Z_{4}Z_{7}$	—	—	—	—	—	$Z_{4}Z_{7}$	—
CNOT error	$Z_{6}^{CT}$	—	—	—	—	—	$Z_{2}^{T}=Z_{5}^{CT}$	—
qubit error	$Z_{2}Z_{4}Z_{6}$	—	—	—	—	—	$Z_{2}Z_{4}Z_{5}$	—

**Table 3 entropy-24-01107-t003:** The relation of possible errors of CNOT gates, the according qubit errors they generated, and the syndromes of both the “flag” qubits and Z stabilizers. Each row indicates the errors with the same “flag” syndrome, and each column is the errors with the same stabilizer syndrome.

*S_*1*Z_, S_*2*Z_, S_*3*Z_*		000	001	010	011	100	101	110	111
*flag*_1_, *flag*_2_, *flag*_3_									
000	CNOT error	-	-	-	$X_{6}^{T}=X_{8}^{T}$	$X_{1}^{T}=X_{4}^{T}$	$X_{2}^{T}=X_{5}^{T}=X_{7}^{T}$	-	$X_{3}^{T}=X_{9}^{T}$
	qubit error	-	-	-	$X_{6}$	$X_{1}$	$X_{5}$	-	$X_{7}$
001	CNOT error	$X_{7}^{CT}$	$X_{9}^{C}$	-	-	-	$X_{7}^{C}=X_{8}^{CT}$	$X_{8}^{C}=X_{9}^{CT}$	-
	qubit error	$X_{4}X_{5}X_{6}X_{7}$	$X_{4}$	-	-	-	$X_{4}X_{6}X_{7}$	$X_{4}X_{7}$	-
010	CNOT error	$X_{4}^{CT}$	$X_{5}^{C}=X_{6}^{CT}$	$X_{6}^{C}$	-	$X_{4}^{C}=X_{5}^{CT}$	-	-	-
	qubit error	$X_{1}X_{2}X_{5}X_{6}$	$X_{2}X_{6}$	$X_{2}$	-	$X_{2}X_{5}X_{6}$	-	-	-
100	CNOT error	$X_{1}^{CT}$	$X_{2}^{C}=X_{3}^{CT}$	-	-	$X_{1}^{C}=X_{2}^{CT}$	-	$X_{3}^{C}$	-
	qubit error	$X_{1}X_{3}X_{5}X_{7}$	$X_{3}X_{7}$	-	-	$X_{3}X_{5}X_{7}$	-	$X_{3}$	-

**Table 4 entropy-24-01107-t004:** The relation of possible errors of CNOT gates, the according qubit errors they generated, and the syndromes of both the “flag” qubits and *X* stabilizers. Each row indicates the errors with same “flag” syndromes, and each column is the errors with the same stabilizer syndromes.

*S_*1*X_, S_*2*X_, S_*3*X_*		000	001	010	011	100	101	110	111
*flag*_4_, *flag*_5_, *flag*_6_, *flag*_7_									
0000	CNOT error	-	$Z_{7}^{C}=Z_{8}^{C}=Z_{9}^{C}$	$Z_{4}^{C}=Z_{5}^{C}=Z_{6}^{C}$	-	-	-	$Z_{1}^{C}=Z_{2}^{C}=Z_{3}^{C}$	-
	qubit error	-	$Z_{4}$	$Z_{2}$	-	-	-	$Z_{3}$	-
0001	CNOT error	$Z_{3}^{CT}$	-	-	-	-	-	$Z_{3}^{T},Z_{9}^{CT}$	$Z_{9}^{T}$
	qubit error	$Z_{3}Z_{4}Z_{7}$	-	-	-	-	-	$Z_{4}Z_{7}$	$Z_{7}$
0010	CNOT error	$Z_{6}^{CT}$	-	$Z_{6}^{T}=Z_{8}^{CT}$	$Z_{8}^{T}$	-	-	-	-
	qubit error	$Z_{2}Z_{4}Z_{6}$	-	$Z_{4}Z_{6}$	$Z_{6}$	-	-	-	-
0100	CNOT error	$Z_{2}^{CT}$	-	-	-	$Z_{5}^{T}=Z_{7}^{CT}$	$Z_{7}^{T}$	$Z_{2}^{T}=Z_{5}^{CT}$	-
	qubit error	$Z_{2}Z_{3}Z_{4}Z_{5}$	-	-	-	$Z_{4}Z_{5}$	$Z_{5}$	$Z_{2}Z_{4}Z_{5}$	-
1000	CNOT error	$Z_{1}^{CT}$	-	-	-	$Z_{4}^{T}$	-	$Z_{1}^{T}=Z_{4}^{CT}$	-
	qubit error	$Z_{1}Z_{2}Z_{3}$	-	-	-	$Z_{1}$	-	$Z_{1}Z_{2}$	-

## Data Availability

Not applicable.

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
