# Peer review of "Implementation of Fault-Tolerant Encoding Circuit Based on Stabilizer Implementation and “Flag” Bits in Steane Code"

_entropy, 2022, doi:10.3390/e24081107_

Round 1

Reviewer 1 Report

Ref.: 1831714

In this work, the authors propose a fault-tolerant enconding circuit based on both stabilizer implementation and flag qubits in the well-known Steane code. Steane code is an important achievement of quantum error correction in quantum computation. Basicaly, Steane code is an extension of the classical Hamming code to the quantum realm and encodes one logical qubit into seven qubits. Steane code is used to correct bit flip (X errors) and phase flip (Z errors) errors that can randomly occur in the encoded qubit. However, extra auxiliary qubits must be added to the circuit in order to perform fault-tolerant error correction. In the circuits proposed in the manuscript, three or four extra qubits are necessary to unambigously distinguish syndromes. The authors simulate such circuits and compare their performance with classical Steane circuit to reduce errors rates.

Quantum computing is no longer just an academic subject. Nowadays, companies and governments are investing resources in developing platforms with large scale quantum processors with dozens or hundreds of qubits. Since qubits are very vulnerable to their environment, quantum error correction is an alternative to circumvent flaws when running quantum algorithms. This fact justifies the relevance of the work. However, it seems to me that the manuscript is not ready to be published. The text contains a lot of misprints or , such as "enviorment", "improvment", "endcoding", "more over", "further more", "analysis" used as a verb, to cite a few.  There are a profusion of the article "the" which, even though it cannot be considered a grammatical error, is not a good practice in English writing (specially in scientific texts). Other constructions are strange, as the ones that combine "while/as" with "so".

Besides the language mistakes pointed out above, the authors must pay attention to other questions.
1. In the penultimate paragraph of Introduction, the work of Chao and Reichardt (reference [25] of the manuscript) is cited and the authors use the idea of flag qubits presented there to develop their proposal. However, the authors do not establish a clear distinction between their proposal and that of Chao and Reichardt that also study a flagged procedure in [[7,1,3]] code corrections. Such a distinction would help to reinforce the relevance of the work.
2. H gate, and H operation are used along the text but the name Hadamard is only mentioned in page 13. I agree that such terms are familiar to readers specialized in quantum circuits or quantum information, however, I think it
is reasonable to indicate what they mean at the first time they appear in the text.
3. In the last paragraphs of Section 5, lowercase "p" and uppercase "P" symbols are used, but only the first was defined as the probability of an error to occur in a quantum gate. What does the uppercase "P" mean?
4. The authors claim that even an error occuring with a given probability in every gate, the proposed circuit has also a certain probability of error, but the error rate has been EXPONENTIALLY reduced compared with the error rate of the original circuit. I do not see how their results sustain such a claim. In fact, compared with the original circuit, their proposal roughly improves the error rates by a factor of 10. So, what does the term "exponentially" mean in this context?  

In my opinion, the ms does not deserve to be published, mainly because it presents a lot of grammatical errors and bad English writing, besides other questions that the authors must pay attention.

Reviewer 2 Report

The manuscript proposes a scheme for the fault-tolerant preparation of the encoded state using flag qubit. The flag qubit techniques have attracted more attention to detect high-weight errors in the encoded state. The conventional scheme with flag qubits employs the ancilla qubits to prepare the encoded state. The authors proposes a scheme that does not employ the ancilla qubits for the encoded state preparation with flag qubits, where flag qubits are interacted with data qubits. Since I think this work is of broad interest to researchers in the field of fault-tolerant quantum computation since the elimination of ancilla qubits in flagged encoding makes flag qubit techniques more attractive. Overall, this manuscript is clear and well organized, and the technical content is understandable, clearly showing their scheme. Thus, I recommend the publication of this manuscript after considering the optional (minor) change.

1.  There have been various proposals for the encoded state preparation with flag qubit techniques, e.g. Refs[25,34,35,36] in the manuscript. I suggest that the authors should explain the difference between the proposed scheme and those more clearly.

Round 2

Reviewer 1 Report

The authors fulfilled the recommendations of my first report. I think the manuscript can be published after minor changes. For example, on page 4, the phrase "...Hadamard gate will be abbreviated as H gate for the rest of this article" should be enclosed in parentheses. In some places, math symbols do not appear in italics. Authors do not use a standard style for citation, for example, they use "et al." to refer to the work of Chao and Reichardt (Ref. [25]) and this work has only two authors. They should also pay attention to the References section to standardize the list of references.